# Ensemble dimensionality reduction and feature gene extraction for single-cell RNA-seq data

Xiaoxiao Sun[1,3], Yiwen Liu [1,3] & Lingling An [1,2,3 ✉]

Single-cell RNA sequencing (scRNA-seq) technologies allow researchers to uncover the biological states of a single cell at high resolution. For computational efficiency and easy visualization, dimensionality reduction is necessary to capture gene expression patterns in low-dimensional space. Here we propose an ensemble method for simultaneous dimensionality reduction and feature gene extraction (EDGE) of scRNA-seq data. Different from existing dimensionality reduction techniques, the proposed method implements an ensemble learning scheme that utilizes massive weak learners for an accurate similarity search. Based on the similarity matrix constructed by those weak learners, the low-dimensional embedding of the data is estimated and optimized through spectral embedding and stochastic gradient descent. Comprehensive simulation and empirical studies show that EDGE is well suited for searching for meaningful organization of cells, detecting rare cell types, and identifying essential feature genes associated with certain cell types.

[1] Department of Epidemiology and Biostatistics, University of Arizona, Tucson, AZ 85724, USA. [2] Department of Biosystems Engineering, University of Arizona, Tucson, AZ 85721, USA. [3] These authors contributed equally: Xiaoxiao Sun, Yiwen Liu. ✉email: anling@arizona.edu

The advent of massive single-cell transcriptomic data provides unprecedented opportunities to study cellular heterogeneity within complex tissues[1–3]. A crucial component for single-cell RNA sequencing (scRNA-seq) data analysis is dimensionality reduction of the large-scale and feature-rich datasets[4,5]. The dimensionality reduction methods consist of two major types, linear and nonlinear techniques. The former one, such as principal component analysis (PCA), has the issue of overcrowding representation for the scRNA-seq data[6,7]. Therefore, nonlinear dimensionality reduction methods such as t-distributed stochastic neighborhood embedding (t-SNE) and uniform manifold approximation and projection (UMAP) are widely used in the scRNA-seq data analysis[8–12]. The t-SNE method is powerful in preserving the local structure of the scRNA-seq data. However, it may suffer from losing the global geometry pattern/property of the data. When the hierarchical structure presents in the data, t-SNE may not be able to capture such a global structure[7]. The UMAP method has been developed to address those issues[12]. It preserves both the local and global structures of cell populations and is more efficient in computation. However, UMAP may be inefficient in identifying rare cell types when dominant/common cell types exist. Our simulated and empirical studies demonstrate that UMAP is less efficient in separating rare cell types from dominant ones and preserving locally hierarchical structures. More importantly, these techniques are not designed to detect feature (i.e., differentially expressed) genes that are associated with various cell types while performing dimensionality reduction.

In this paper, we propose an Ensemble Dimensionality reduction and feature Gene Extraction (EDGE) method to simultaneously perform dimensionality reduction and feature gene identification. Our method adopts an ensemble learning technique, which employs multiple weak learners to obtain better predictive performance than could be obtained from any weak learners alone[13]. Thus, massive weak learners are allowed in EDGE to learn cell similarities accurately, while the genes that make significant contributions in the process are selected as feature genes. A series of comprehensive simulation and empirical studies demonstrate the high effectiveness of EDGE in identifying rare cell types, preserving local and global structures, and detecting vital feature genes.

## Results

**Overview of EDGE.** To conduct dimensionality reduction, it is pivotal to construct the similarity/dissimilarity matrix between cells faithfully. Similarity learning for massive amounts of feature-rich data emerges as a challenging problem. Several methods have been proposed to learn the similarity[14–16]. Nevertheless, similarity learning in the scRNA-seq data, by its nature, is complicated. First, traditional methods only work well when the dimensionality is relatively low[17,18], thus are not applicable to feature-rich scRNA-seq data. Second, due to the dropout effects, it is even more challenging to obtain accurate similarity measures[19]. Third, the existence of rare cell groups and/or subgroups adds another layer of complications. Owing to the data heterogeneity, rare cell groups often get overlooked, while cell subtypes are usually masked or nested within upper-level cell types[20].

To address the above concerns, we propose to use EDGE to learn the similarity between cells. The proposed approach is motivated by a similarity search method, sketching[21,22]. Jindal et al.[20] proposed the finder of rare entities (FiRE) algorithm based on the sketching technique. It successfully identified rare cell types in a matter of seconds. Although motivated by the same sketching technique, EDGE is entirely different from FiRE in three aspects. First, we combine ensemble learning with the sketching technique to calculate the similarity probabilities between cells based on the gene expression matrix. The sketching technique estimates similarity scores between cells using the sketch (bit vector) in each weak learner. Regardless of the dropout events, those bit vectors are capable of capturing the information contained in nonzero values with higher probability. Following the scheme of ensemble learning, the final similarity probabilities between cells are the averaged similarity scores from each weak learner (Fig. 1). Second, EDGE uses the similarity probabilities to

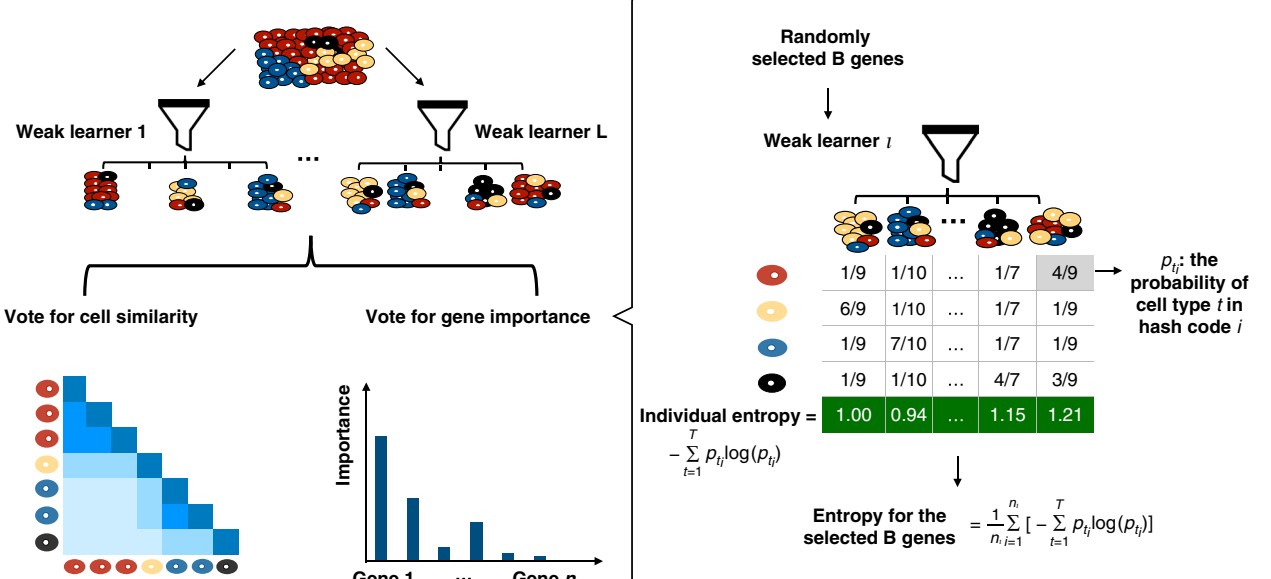

**Fig. 1 Overview of EDGE.** The algorithm starts by generating a number of weak learners. Each weak learner consists of a few hash codes (imaginary boxes, i.e., piles of cells). For cells assigned to the same hash code, their pairwise similarity scores are set to be 1s and 0s otherwise. Each weak learner is a voter. The final similarity probabilities between cells are calculated by averaging the corresponding similarity scores from each voter. The calculated probabilities are used in embedding estimation and optimization. The important scores of genes in each weak learner are obtained by averaging the hash codes' entropy values. Details of the algorithm can be found in Methods and Supplementary Information.

learn and optimize the low-dimensional embedding through minimizing a cross-entropy function by stochastic gradient descent[12,23]. Last, the proposed method can detect feature genes that contribute to the separation of various cell types. The detection of feature genes relies on trained weak learners in the first step. To this end, EDGE uses the ensemble learning to construct the low-dimensional embedding and perform feature gene selection simultaneously. A visual illustration of the EDGE method is provided in Fig. 1, and an elaborate explanation of the EDGE method is presented in the "Methods" section.

**Benchmarking EDGE in simulated studies.** We designed simulated experiments to evaluate the performance of EDGE in embedding the synthetic scRNA-seq count data with the presence of rare cell types and at various rates of dropout events. The proposed method was compared with t-SNE and UMAP, which were widely used dimensionality reduction methods in scRNA-seq data analysis[24–26]. We generated the simulated data using Splatter under four scenarios (Methods)[27]. Scenario 1: One thousand cells in four groups, with the ratio of 10:10:10:70 in group size (number of cells) and a low dropout rate in gene expression of 1,500 genes; Scenario 2: Four equal-sized groups of cells, i.e., each with a proportion of 25% of the total 1000 cells, and gene expression of 1500 genes at a high dropout rate; Scenario 3: Four groups of 1000 cells, with the ratio of 10:10:10:70 in group size and a high dropout rate in gene expression of 1500 genes; Scenario 4: Four equal-sized groups of 1000 cells with gene expression of 1500 genes at a low dropout rate. The proportions of differentially expressed (DE) genes in Scenario 1 to 3 were fixed at 35%, while varied from 10% to 25% in Scenario 4. The observed zero proportions for the high and low dropout rates are around 0.70 and 0.87. The details of the simulation settings are shown in Table 1 and Methods. Examples of embedding figures for Scenario 1 and 2 are shown in Supplementary Fig. 1.

To investigate the abilities of EDGE in preserving the structure of cell populations, we trained random forests to predict cell clusters' identities using the embeddings generated by the aforementioned methods for comparison[12,28]. The within-group and overall accuracy of the predictions were measured through out-of-bag (OOB) prediction errors over 100 simulation replicates. Furthermore, we reported the median of silhouette scores, denoted by the silhouette index, for each embedding to evaluate the performance of embedding methods[29,30]. The range of the silhouette index is from −1 to +1, and the positive one represents well-separated clustering. When the dropout rates were around 0.7 in the presence of rare cell types (Fig. 2a), EDGE led to higher prediction accuracy, especially for rare cell types. For the case of equal group size, even with high zero proportion (close to 0.9), EDGE appeared more efficient than the other two methods at separating cell types (Fig. 2b). When there existed rare cell types,

coupled with the presence of high dropout rates (Fig. 2c), all methods became less efficient, but EDGE still ranked first in terms of prediction accuracy. When the proportions of rare cell types further decreased to 0.1% or the number of cell types varied, EDGE still outperformed other methods in predicting rare cell types (Supplementary Figs. 2–5). We also evaluated how reliable the embedding was when the proportions of DE genes varied (Scenario 4). EDGE and UMAP achieved higher overall prediction accuracy, followed by t-SNE (Fig. 2d). The silhouette index was consistent with the prediction accuracy of random forests (Fig. 2a–d). EDGE achieved a higher silhouette index on average in almost all the scenarios. In summary, the results suggest that the embedding generated by EDGE accurately characterizes all cell groups and is effective in separating rare cell types in different scenarios.

We also evaluated the performance of EDGE for detecting the feature (DE) genes in simulated studies. Since the t-SNE and UMAP methods were not designed to identify feature genes, we presented the performance of EDGE in identifying true feature genes. Owing to the multi-layer data generation procedure in Splatter, the true feature genes set at the first stage may not be differentially expressed in the final simulated data[27]. We thus modified the procedure (Methods) and simulated scRNA-seq datasets in four scenarios. In the first two scenarios, two types of 1000 cells with proportions of about 80% and 20% were generated. Among the total of 500 genes, 30 of them were set as true feature genes in each two-group scenario. We repeated the simulation 100 times and reported results in Table 2. The means of zero proportions over 100 replications for the high and low dropout scenarios were 78.83% and 50.55%. The proposed algorithm has a high accuracy rate in detecting true feature genes. In the low dropout scenario, when the top 15 genes were chosen based on the importance scores, all of them were the true feature genes; and when we selected the top 30 genes, 27.06 on average were. In the high dropout scenario, although the signal-to-noise ratio decreased, we still detected 14.79 and 25.74 true feature genes on average. The simulation results for three cell types are shown in Supplementary Table 1.

**EDGE is accurate in embedding rare cell types.** To investigate the performance of EDGE in handling rare cell populations, we applied EDGE alongside t-SNE and UMAP to two scRNA-seq datasets (the Jurkat dataset and the Peripheral Blood Mononuclear Cell (PBMC) dataset). The Jurkat dataset contains two cell types, 293T and Jurkat cells (Methods), and the proportion of Jurkat cells is around 2.5%[20,31]. The PBMC dataset contains nine cell types, among which four are rare, with the concentration varying from 0.89% to 2.2% (Methods). For each dataset, we projected cells into a two-dimensional space based on the embeddings generated by the three methods. Figure 3 demonstrates the performances of those methods in separating rare cell types from the other cell types. EDGE captured much information in the Jurkat dataset and outperformed t-SNE and UMAP (Fig. 3a–c). It successfully separated all Jurkat cells (the rare cell type, in red) from 293T cells (the dominant type, in blue) (Fig. 3a). However, t-SNE and UMAP mapped some 293T cells to the cluster of rare cells (Fig. 3b, c). In analyzing the PBMC dataset, the four rare cell types were separated into more compact clusters with clearer patterns by EDGE than by t-SNE and UMAP. For instance, dendritic and megakaryocyte cells formed distinct clusters based on the low-dimensional representation of EDGE (in yellow and brown, respectively, Fig. 3d). Nonetheless, these cells were close to other cell types in the low-dimensional subspace generated by t-SNE and UMAP (Fig. 3e, f). In addition, plasmacytoid dendritic cells (in gray)

**Table 1 Settings in simulation studies.**

| | Group proportions (%) | Zero proportion | DE gene proportion (%) |
|---|---|---|---|
| Scenario 1 | (10, 10, 10, 70) | 0.6993 (0.0229) | 35 |
| Scenario 2 | (25, 25, 25, 25) | 0.8682 (0.0169) | 35 |
| Scenario 3 | (10, 10, 10, 70) | 0.8688 (0.0165) | 35 |
| Scenario 4 | (25, 25, 25, 25) | 0.6961 (0.0233) | 10 |
| | | 0.6965 (0.0234) | 15 |
| | | 0.6968 (0.0231) | 20 |
| | | 0.6974 (0.0233) | 25 |

The standard deviation of the observed zero proportions in simulated data over 100 replications is shown in the parentheses.

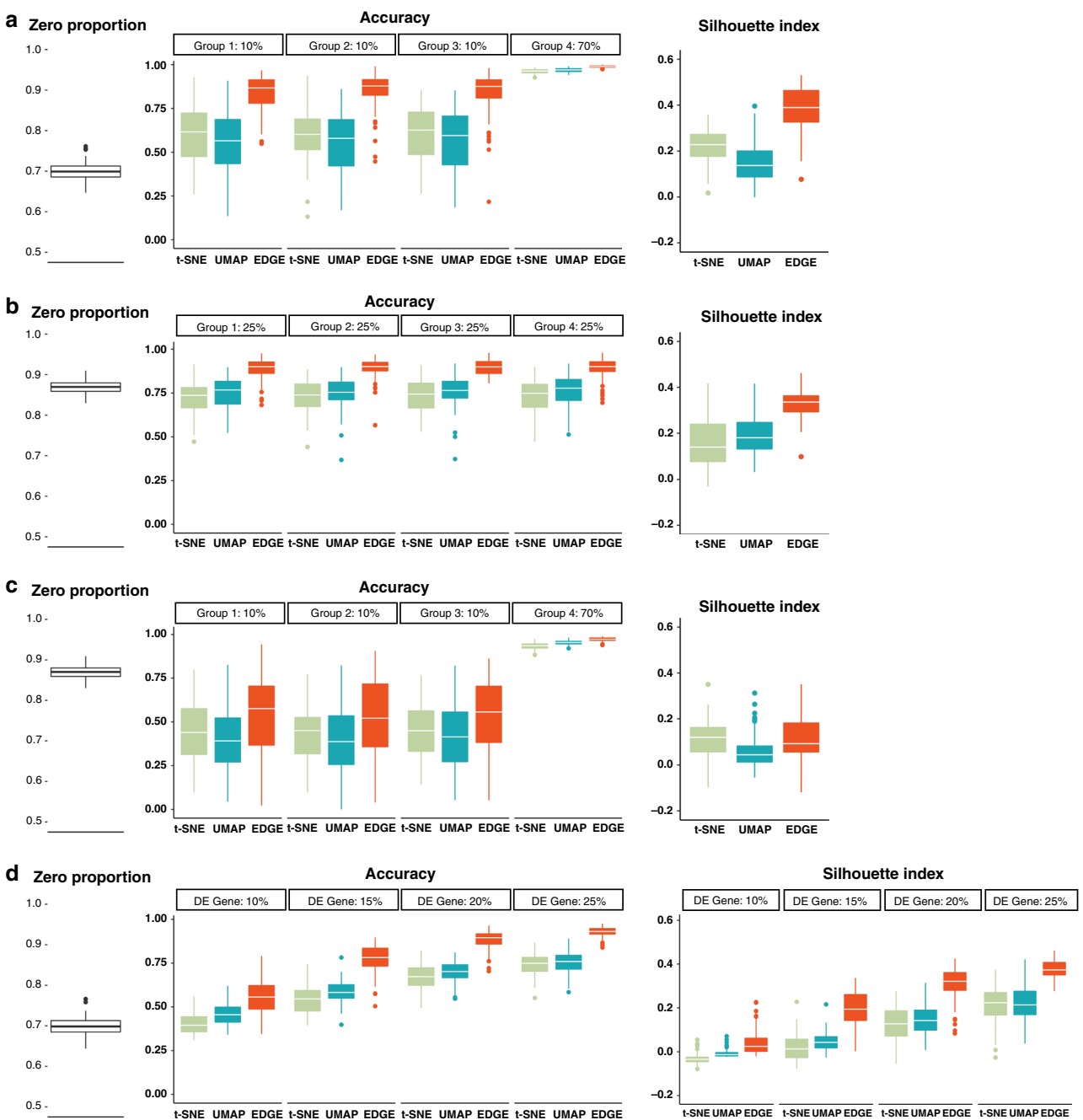

**Fig. 2 EDGE preserves the structure of cell populations.** The results over 100 replications of simulation studies are shown in **a** (Scenario 1), **b** (Scenario 2), **c** (Scenario 3), and **d** (Scenario 4). In each panel, the left figure presents observed zero proportions of the simulated scRNA-seq datasets; the middle figure shows the accuracy of random forests in predicting cell cluster labels using the learnt embeddings as input; the right figure displays the calculated silhouette indices of embeddings. In the boxplots, we show the median (central lines), first and third quartile (box limits), and the whiskers extended to the lowest and highest points within 1.5 interquartile range of the first and third quartiles, respectively.

**Table 2 The performance of EDGE in detecting the feature genes in simulated studies with two cell types over 100 replications.**

|  | Top 15 genes | Top 30 genes |
| --- | --- | --- |
| Low dropout | 15.00 (0.00) | 27.06 (1.38) |
| High dropout | 14.79 (1.46) | 25.74 (3.03) |

The standard deviation of the number of identified true feature genes is shown in the parentheses.

formed a distinct cluster in all low-dimensional subspaces. However, some B cells (in red) were mapped to the corresponding regions in the two-dimensional scatter plots of t-SNE (Fig. 3f). We also calculated the silhouette indices of the embeddings estimated by the methods for comparison over 100 replications. The mean (standard deviation) of the silhouette indices of EDGE for the four rare cell types was 0.851(0.045), whereas the values for UMAP and t-SNE were 0.801(0.012) and 0.838(0.008), respectively. Taken together, these analyses demonstrate that EDGE is an accurate method in separating rare cell types compared to t-SNE and UMAP.

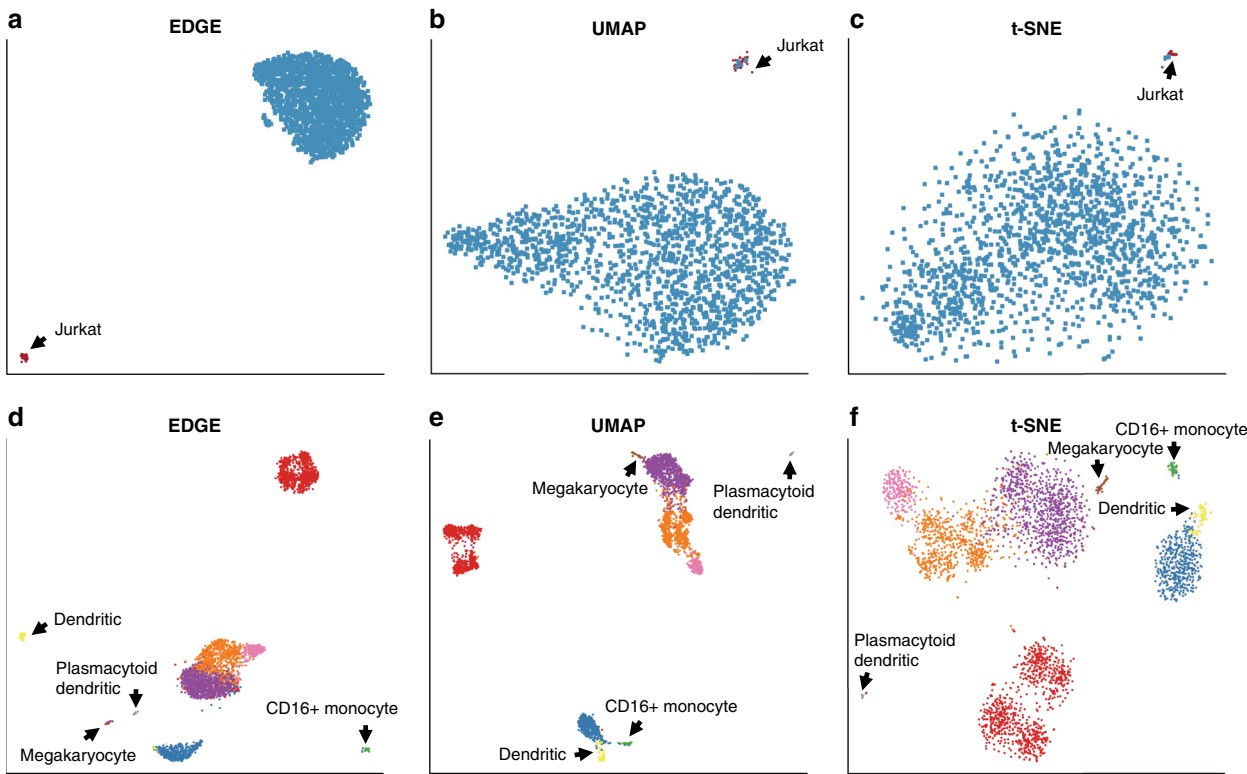

**Fig. 3 Detection of rare cell types. a–c** The two-dimensional embeddings learnt by EDGE, UMAP, and t-SNE, respectively, on the Jurkat dataset. **d–f** The two-dimensional embeddings learnt by EDGE, UMAP, and t-SNE, respectively, on the PBMC dataset. Black arrows annotate rare cell types.

**EDGE better preserves global and local structures**. We applied EDGE, t-SNE, and UMAP to the mouse brain dataset[32] and illustrated that EDGE better preserved global and local structures. The dataset consists of 3005 cells in the mouse cortex and hippocampus (Methods). We first fitted a 30-dimensional ellipsoid to the dataset using PCA. Then cells in the 30-dimensional subspace were projected onto a two-dimensional space generated by the embeddings from the three methods. As demonstrated in Fig. 4, seven types of cells identified in the original study were mapped to seven compact regions in the EDGE-based two-dimensional subspace (Fig. 4a). Compared to EDGE, some clusters were scattered in the two-dimensional subspace estimated by t-SNE and UMAP (Fig. 4b, c). We calculated silhouette indices of the estimated embeddings by three methods over 100 replications. The means (standard deviations) of silhouette indices of all cell types were 0.621(0.085) for EDGE, 0.597(0.025) for UMAP, and 0.501(0.017) for t-SNE. In addition to preserving the global structure, EDGE showed promising results in maintaining hierarchical structures of cells. It mapped cell subtypes within the same cell type together in the low-dimensional space. For instance, five subtypes of astrocytes-ependymal cells were present within a compact region annotated by the black arrow (Fig. 4d). However, in the resulting plots of UMAP and t-SNE, one subtype (annotated by the black arrow) was mapped to a region far away from the region of other subtypes (Fig. 4e, f). Similarly, four subtypes of microglia cells were also projected to a clearly distinguishable region in the EDGE-based two-dimensional subspace (Fig. 4g), whereas those four cell subtypes were mapped to different regions in the UMAP-based and t-SNE-based subspaces (Fig. 4h, i).

**EDGE is capable of detecting feature genes**. EDGE is able to identify feature genes that are responsible for the separation of different cell populations. It ranks all candidate genes according to their contribution of predicting cell identities, which is referred to as the importance score. Then the number of feature genes is determined by the distribution of those importance scores (Methods). We identified 17, 35, and 43 feature genes for the Jurkat, PBMC, and mouse brain datasets (Supplementary Data 1). In the 35 detected genes in PBMC dataset, *PPBP*, *LYZ*, *CST3*, and *NKG7* were the marker genes for platelet, CD14+ monocyte, dendritic cells, and natural killer cells, respectively (Fig. 5)[33]. Feature genes detected by EDGE were classified into two types. For the first type, genes such as *ACRBP* and *IGLC3* were solely expressed in a specific cell type. This type of genes was also detected in the Jurkat dataset (Supplementary Fig. 6). Such genes could be identified using standard methods, e.g., fold change[34]. Genes of the second type separated different cell types based on their unique distribution patterns of gene expression values in some cell types. For instance, the most important gene *S100A9* (leftmost gene in Fig. 5) was highly expressed in CD14+ monocyte, CD16+ monocyte, and dendritic cells. While this gene distinguished these three cell types from the remaining, the unique distribution patterns of expression levels in these three cell types (violin shapes in Fig. 5) were beneficial to further differentiate three of them. These two types of genes were also found in the mouse brain dataset (Supplementary Fig. 7), for example, *MPB* and *ARAP3*.

Furthermore, we performed gene ontology (GO) enrichment analysis for the 35 detected genes in PBMC dataset[35,36] and showed ten most enriched GO biological processes in Table 3. All ten enriched biological processes were related to immune response and response to stimulus. Since PBMC cells such as B cells and T cells initiated or got involved in immune responses, the enriched biological processes were highly correlated with the biological functions of PBMC cells[37].

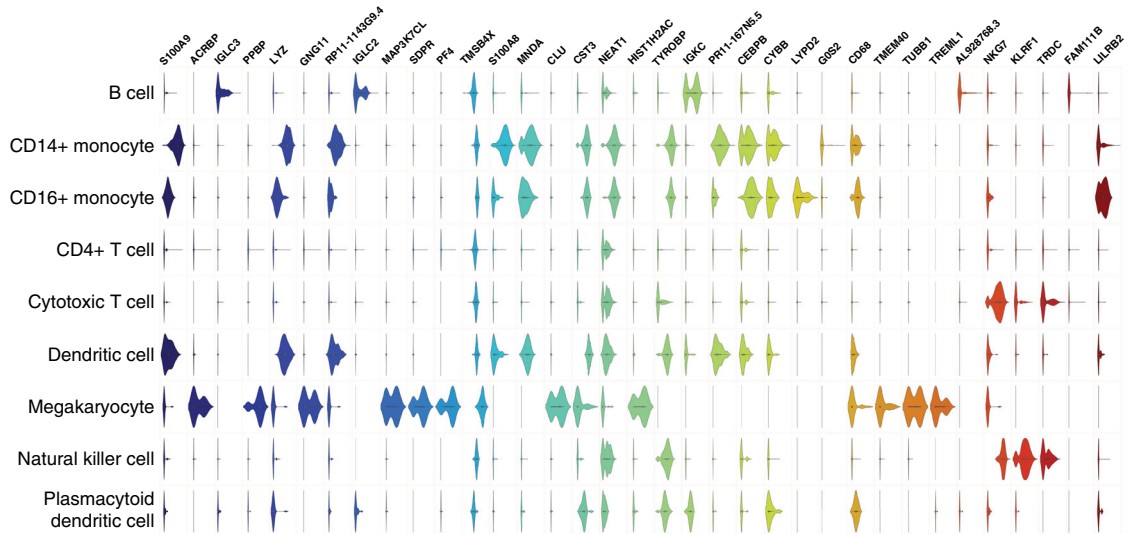

**Fig. 4 Local and global structure can be preserved by EDGE in the mouse brain dataset. a–c** The two-dimensional representations learnt by EDGE, UMAP, and t-SNE, respectively. **d–f** Same as **a–c**, but only the astrocytes-ependymal cells are highlighted with their subtypes. **g–i** Same as **a–c**, but only the microglia cells are highlighted with their subtypes.

**Fig. 5 Normalized expression levels of 35 top-ranked feature genes detected by EDGE for the PBMC dataset.** Genes are ordered by their importance scores with *S100A9* (the most left on top) having the highest importance score.

**Table 3 Ten most enriched GO biological processes for the PBMC scRNA-seq dataset.**

| GO biological process | Fold enrichment | Raw P-value | FDR |
|---|---|---|---|
| Defense response GO:0006952 Leukocyte-mediated immunity GO:0002443 | 13.81 | 3.52E-15 | 5.60E-11 |
| Immune effector process GO:0002252 | 10.33 | 3.74E-14 | 1.99E-10 |
| Immune response GO:0006955 | 6.67 | 1.30E-12 | 3.45E-09 |
| Response to external biotic stimulus GO:0043207 | 8.37 | 1.09E-12 | 3.47E-09 |
| Response to biotic stimulus GO:0009607 | 8.17 | 1.59E-12 | 3.62E-09 |
| Response to other organism GO:0051707 | 8.38 | 1.06E-12 | 4.24E-09 |
| Defense response to other organism GO:0098542 | 10.22 | 2.50E-12 | 4.97E-09 |
| Humoral immune response GO:0006959 | 19.40 | 5.81E-12 | 1.03E-08 |
| Regulated exocytosis GO:0045055 | 12.27 | 1.23E-11 | 1.96E-08 |

The Fisher's exact test was conducted to obtain the raw P-values, which were further adjusted by Benjamini–Hochberg False-Discovery Rate (FDR) correction.

## Discussion

We have developed an ensemble method, EDGE, for simultaneous dimensionality reduction and feature gene extraction in scRNA-seq data. It exploits a large number of weak learners to study the similarities between cells. Those massive weak learners not only vote for cell similarities but also detect feature genes that contribute the most in the voting. Furthermore, those learners are constructed through the sketching technique, and EDGE thus is sensitive to rare cell types. There are three major contributions of this study. First, EDGE provides an accurate way for similarity search (Supplementary Figs. 8–9). In all three real datasets, EDGE shows more compactly projected structures compared with t-SNE and UMAP (Figs. 3 and 4a–c). Second, EDGE preserves both local and global structures in scRNA-seq data. When there exist rare cell types, EDGE provides a neat separation between rare and common cell types (Fig. 3a, d). For cells that belong to different subtypes, EDGE maintains the hierarchical structures of cell types successfully. Third, compared to popular embedding methods such as t-SNE and UMAP, EDGE can effectively detect feature genes that are crucial for discerning different cell types. Additional discussion about the differences among EDGE, t-SNE, and UMAP can be found in Supplementary discussion. The way we vote for cell similarity and gene importance has a natural connection with the popular ensemble method, random forests. The success of random forests relies on two prerequisites: (1) the decision trees are weak learners, and (2) the predictions made by the decision trees have low correlations[38]. In the proposed algorithm, we build each learner by randomly selecting a group of genes. Since each learner uses different sets of genes, cell similarity scores for the learners are not highly correlated. Besides, the number of selected genes is much smaller than the total number of genes. Thus, each learner is weak in terms of prediction accuracy of cell similarities. Based on these two prerequisites, EDGE are accurate in estimating cell similarities and gene importance scores.

## Methods

**Similarity search.** Our algorithm starts with constructing a similarity learner. We randomly select $B$ genes out of all genes. We then randomly pick a gene-specific threshold within the range of all values of gene expression matrix $\mathbf{X} \in \mathbb{R}^{C \times G}$. For each cell, we calculate a sketch vector $\mathbf{V} \in \{0, 1\}^B$, which is a bit vector with $B$ elements (0 or 1). Each element is associated with a selected gene. If the gene expression value is greater than the gene's threshold, its corresponding value in the bit vector is 1 and 0 otherwise. Let $\mathbf{W} \in \mathbb{R}^B$ be the randomly generated weight vector. We use modulo hashing technique to map $\mathbf{V} \cdot \mathbf{W}$ to one of the predefined hash codes, where $\cdot$ represents dot product. A hash code can be viewed as an imaginary box in which similar cells are stored. The similarity score of cells $i$ and $j$ in the same hash code is set to be 1, i.e., the $(i, j)$th entry of similarity score matrix is 1. Once the pairwise similarity scores for all cells are calculated, the construction of a similarity learner is complete. Owing to the randomness in constructing the similarity learner, it is possible that the similarity score for dissimilar cells is 1. Thus, we call the similarity learner as the weak learner to reflect its accuracy in similarity learning. To increase the accuracy of such similarity learning, we utilize ensemble learning by constructing $L$ weak learners. Each weak learner is a voter. The final similarity matrix $\mathbf{S}$ is calculated by averaging the corresponding similarity scores from all voters,

$$\mathbf{S} = 1/L \sum_{l=1}^{L} \mathbf{S}_l, \tag{1}$$

where $\mathbf{S}_l \in \mathbb{R}^{C \times C}, l = 1, \cdots, L$, is the similarity score matrix in each weak learner. The detailed process is described in Supplementary Algorithm 1.

**Spectral embedding.** The next stage of the proposed method is the construction of a k-nearest neighbor (k-NNG) graph with the weighted adjacency matrix $\mathbf{S}$ in Equation (1). Once the graph is constructed, the spectral embedding is performed on the normalized Laplacian $\mathbf{D}^{1/2}(\mathbf{D} - \mathbf{S})\mathbf{D}^{1/2}$, where $\mathbf{D}$ is the degree matrix for $\mathbf{S}$. The output of this stage is top $d$ eigenvectors of normalized Laplacian, $\mathbf{E}_d$. The detailed process is described in Supplementary Algorithm 2.

**Embedding optimization.** Briefly, the optimization stage keeps similar cells close to each other and dissimilar cells far apart in the low-dimensional space. The optimization algorithm includes two stages in which a stochastic gradient descent algorithm with the decreasing step size is implemented to minimize the loss function. In the first stage, one updates according to the value of $\log(\Phi(\mathbf{e}_m, \mathbf{e}_n))$, where $\Phi(\mathbf{e}_m, \mathbf{e}_n) = (1 + a(\|\mathbf{e}_m - \mathbf{e}_n\|_2^2)^b)^{-1}$, $\mathbf{e}$. is the eigenvector of the cell $\cdot$ in $\mathbf{E}_d$, $n$ is the nearest neighbor of $m$ based on the k-NNG graph. The hyperparameters $a$ and $b$ are estimated by nonlinear least squares in ref. [39]. In the second stage, one updates according to the value of $\log(1 - \Phi(\mathbf{e}_m, \mathbf{e}_o))$, where $o$ is one of negative samples, i.e., membership strength is 0. For sufficiently large samples, the negative samples are randomly selected using a uniform distribution.

**Feature gene selection.** A two-stage algorithm is implemented to identify feature genes. In the first stage, we apply a shared nearest neighbor modularity optimization-based clustering algorithm to the optimized embedding matrix[33,40]. The predicted labels of cells are the input of the second stage, in which we select feature genes based on the measure of information entropy. Particularly, the entropy for weak learner $l$ based on randomly selected $B$ genes is measured by

$$\frac{1}{n_l} \sum_{i=1}^{n_l} \left( -\sum_{t=1}^{T} p_{t_i} \log p_{t_i} \right), \tag{2}$$

where $n_l$ is the number of hash codes, $T$ is the number of cell types predicted in the first stage, and $p_{t_i}$ is the proportion of cell type $t$ at the $i$th hash code. We then assign the values of entropy in (2) to the selected $B$ genes. The genes that are randomly picked up in different weak learners are varied. With enough $L$, every gene should be selected at least one time. The averaged entropy values over $L$ weak learners for $G$ genes are used to identify feature genes. Let $\mu$ and $\sigma$ be the mean and standard deviation of the averaged entropy values over $L$ weak learners. We choose $\mu - 1.5 \times \sigma$ as the cutoff value to select top feature genes.

**Hyperparameters.** The EDGE algorithm takes five important hyperparameters:

- $L$: the number of weak learners;
- $H$: the hash table size;
- $B$: the number of genes to construct weak learners;
- $k$: the number of nearest neighbors; and
- $d$: the number of eigenvectors.

The number of weak learners, $L$, represents some degree of trade-off between low variance estimation and high computational cost. The default value for $L$ in our

algorithm is 500 (Supplementary Figs. 10 and 12). The hash table size, i.e., $H$, is set 1,017,881 for all datasets[20]. A large $B$ makes the algorithm sensitive to noises. Thus, values for $B$ are typically <0.05*$G$. The EDGE method is robust to the number of nearest neighbors, $k$. The typical values are from 10 to 50 (Supplementary Figs. 11 and 12). The dimensionality $d$ is usually set to be two for visualization or be the number of target clusters plus one for prediction.

In real datasets, the default parameters were used for the t-SNE and UMAP methods except for the number of nearest neighbors. For a fair comparison, we utilized the same number of nearest neighbors for EDGE and UMAP.

**Computational complexities.** One advantage of the sketching method is that it is computationally efficient and can build the learners in linear, i.e., $O(C)$, time[20]. For optimization of embedding, the computational complexity is $O(kC)$[41,42]. The most time-consuming stage in our algorithm is spectral embedding. To significantly reduce the computational burden in this step, we use the RSpectra package to perform the large-scale eigenvalue decomposition[43]. The user time of EDGE for the varied number of weak learners was shown in Supplementary Fig. 12. It took <5 s when 500 weak learners were constructed for datasets with 1000 cells and 500 genes. Note that all weak learners are constructed independently. It is thus convenient to implement our method under the parallel computing framework, which could make EDGE much faster.

**Simulation studies.** We utilized the R package Splatter to simulate scRNA-seq datasets[27]. In all scenarios for dimensionality reduction, we generated 1000 cells and 1500 genes per cell and varied the proportion of differentially expressed genes, the group structure of cell populations, and the dropout rate.

To simulate scRNA-seq data for feature gene identification by EDGE, we first simulated single population scRNA-seq data with 1000 cells and 500 genes. Secondly, cell type labels were randomly assigned to the cells. For the scenarios with two cell types, the ratio of two cell types was 80:20, whereas the ratio, 30:30:40, was used in the scenarios with three cell types. Lastly, we converted nonzero values to zeros for the feature genes in the control group.

**Data preprocessing.** We normalized simulated and real datasets using the median normalization method[20]. The normalized datasets were then $\log_2$ transformed after the addition of one. We selected 1000 top variable genes for real datasets and 500 for simulated datasets using the variance of standardized values, which were calculated by the FindVariableFeatures function in the R package Seurat[33].

**Reporting summary.** Further information on research design is available in the Nature Research Reporting Summary linked to this article.

## Data availability
Below we describe all of the real scRNA-seq datasets used in the current study. All datasets are publicly available and well-studied. The original Jurkat dataset contains about 3200 cells and the expression of 32,738 genes. It is available from https://support.10xgenomics.com/single-cell-gene-expression/datasets. The unique high-quality single nucleotide variants (SNVs) observed in each cell type were used to resolve the cell types. These two types of cells (Jurkat and 293T) are mixed at the ratio of 50:50[31]. To have the rare cell phenomenon, we used the dataset with the Jurkat cell proportion of ~2.5%[20]. The PBMC dataset consists of 3362 cells and 33,694 genes sequenced by the 10x Chromium method[44]. It is available from https://singlecell.broadinstitute.org/single_cell/study/SCP424/single-cell-comparison-pbmc-data. The Cell Ranger pipeline (v2.0.0) was used to process the PBMC dataset. Nine cell types were detected based on known marker genes. For the mouse brain dataset, there are 19,972 genes in 3005 cells[32]. Seven major cell types and 47 molecularly subtypes were identified by the BackSPIN algorithm developed by authors of the original paper. The results were further verified by the authors using known marker genes. The mouse brain dataset is available from https://storage.googleapis.com/linnarsson-lab-www-blobs/blobs/cortex/expression_mRNA_17-Aug-2014.txt.

## Code availability
The EDGE R package is freely available on GitHub (https://github.com/shawnstat/EDGE).

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

## Acknowledgements

This work has been partially supported by the United States Department of Agriculture (ARZT-1360830-H22-138 and ARZT-1361620-H22-149) to L.A.

## Author contributions

X.S., Y.L., and L.A. contributed equally in conceiving the project. X.S. and Y.L. designed the algorithm and software and performed data analysis. X.S., Y.L., and L.A. wrote and revised the manuscript.

## Competing interests

The authors declare no competing interests.
