## [Peer Review File · Nature Communications]

REVIEWER COMMENTS

Reviewer #1 (Remarks to the Author):

This paper develops a new method named EDGE for dimension reduction in single cell data. The EDGE method contains similarity matrix construction, Spectral embedding, embedding optimization, and feature gene extraction. Low dimension embedding is one of the most important technique in single cell data analysis. Authors have compared EDGE with other methods, and show their method has advantage on organization of cells, detecting rare cell types, and identifying essential feature genes. Detection of rare cell types is one main challenge in this area. If the authors could use well-designed simulation study to show the advantage of rare cell type detection, it would be a great contribution to this field. As a methodology paper, I think authors need to improve on following aspects 1) dissect the EDGE method to show the contribution of each step, and 2) give more comprehensive comparison with existing methods.

Major comments:

- 1, What is the reason of EDGE perform so good? I think it is good to dissect the results step by step to show contribution of each step. Which step is the main reason of the improvement? the Similarity matrix? Spectral embedding? Or Embedding optimization? For example, what happen if you use UMAP and t-SNE on the similarity matrix from EDGE (the output of first step). What happen if you use the similarity matrix used in UMAP to optimize the embedding by EDGE.
- 2, The evaluation of the simulation study is not direct. First, authors need to show the accuracy from random forest is a good evaluation metric. The embedding figures of the simulation study should be presented.
- 3, Both UMAP and tSNE don't always give consistent results from run to run. So comparison on one single run is not very convincing. I think authors should run multiple times and use a quantitate way to compare the results on both simulation data and real data.

Minor comments:

- 1, I suggest authors use more comprehensive simulation study. Simulation study should include minor population ranging from 0.1% to 5%. Goal is to study the minimal detectable percentage of different methods. The number of clusters also have impact on the results. Authors should report how rare population detection accuracy change as the number of clusters increase (i.e. up 10 clusters).
- 2, The original paper of single cell brain data has detected nine clusters. But authors only compare 7 clusters. In addition, it seems the t-SNE in original paper has very good separation. I think authors should compare EDGE results with the t-SNE in original paper.
- 3, The brain data is from Fluidigm C1, which data have much lower drop-out rate compared to drop-seq based methods. It seems EDGE can handle high drop-out sparse data. If this is the case, I think it is good to try this method on more challenging data like single cell chromatin accessibility (scATAC-seq). It's just a suggestion.

Reviewer #2 (Remarks to the Author):

The paper entitled "EDGE: Ensemble Dimensionality Reduction and Feature Gene Extraction for Single-cell RNA-seq Data" proposes a novel method for simultaneous dimension reduction and marker gene detection. The authors demonstrate the method's superior performance in both simulation and empirical study datasets. A visualization method that represents clusters more clearly and can jointly identify marker genes would be very useful in single-cell analysis.

Major:

1. I found some of the method descriptions unclear. These should be addressed:
 - i. A one sentence explanation beyond "similarity search method" on what a 'sketch' is would be

very helpful in the last paragraph on page 3.

ii. What does a 'vote' for cell similarity mean?

iii. What exactly is the weak learner method? It is never described in the methods or in the supplement.

2. To justify the methods claim at preserving separation more accurately, the authors could vary the degree of separation among the clusters in the simulation and then define a measure of cluster distances and show whether the distance changes relatively for the EDGE method.

3. Unless the empirical datasets are both sorted single-cell populations, the methods should discuss how the ground truth labels were obtained originally as they were likely estimated using other statistical methods and so have some degree of uncertainty.

4. The package should be made available on Github with a full working vignette with examples. I was not able to test the package myself since there is no documentation on how to use the various functions.

Minor:

1. On page 5, "Differentially expressed (DE) gene proportions for Scenario 1 to 3 were fixed at 35%," differentially should be differentially.

2. The grammar in figure 1 caption can be cleaned up a bit as well as in the first methods paragraph.

3. It would be more clear/fair to write the UMAP and TSNE were not designed to identify feature genes, in addition to saying they aren't able to on Page 6 bottom paragraph.

Response to the Reviewer 1’s comments on “EDGE: Ensemble Dimensionality Reduction and Feature Gene Extraction for Single-cell RNA-seq Data”

We would like to thank you for your insightful comments and suggestions. We have made every effort to address the concerns raised by you and the other reviewer. We hope you find our responses satisfactory. In what follows, in italics we quote your comments and then give our detailed response. We showed all changes in the manuscript text file with color highlighting (blue).

Major Comments:

1. What is the reason of EDGE perform so good? I think it is good to dissect the results step by step to show contribution of each step. Which step is the main reason of the improvement? the Similarity matrix? Spectral embedding? Or Embedding optimization? For example, what happen if you use UMAP and t-SNE on the similarity matrix from EDGE (the output of first step). What happen if you use the similarity matrix used in UMAP to optimize the embedding by EDGE.

Response: Thank you for this helpful advice. We have added one section, **From t-SNE and UMAP to EDGE** in Supplementary Information, to explain the EDGE method step by step. Below, we provide a summary of this section. EDGE, UMAP, and t-SNE share two major steps. **In the first step**, the similarity probabilities between cells in the high-dimensional space are calculated based on the gene expression matrix. **In the second step**, the similarity probabilities between cells in the low-dimensional space, constructed by the embedding matrix, are estimated by minimizing a loss function. The loss function measures the overall distance between similarity probability distributions in the high- and low-dimensional spaces. Through minimizing it, the similarity probability distribution in the low-dimensional space matches the one in the high-dimensional space as close as possible.

The first step is critical in dimensionality reduction. Suppose the similarity probabilities in the high-dimensional space cannot preserve the similarity structures between cells

faithfully, it is impossible to obtain an accurate embedding matrix. EDGE is better at preserving such structures compared to UMAP and t-SNE. To graphically illustrate the advantages of our algorithm in this step, we generated scRNA-seq data containing two cell types with a ratio of 98:2 using Splatter. We then calculated the similarity probabilities in the high-dimensional space using algorithms of EDGE, UMAP, and t-SNE, respectively. The same embedding procedure (spectral embedding) was implemented to map these probabilities onto a two-dimensional space. In **Figure S6**, major and rare cells were well-separated in the two-dimensional space construed by EDGE, whereas some of these cells were mixed in the low-dimensional space learnt by UMAP and t-SNE. We also replaced the similarity probabilities for UMAP and t-SNE with the ones from EDGE. In the embedding spaces based on our similarity probabilities, major and rare cells were mapped to different regions perfectly (**right panel of Figure S7**). For embedding results using original similarity probabilities from UMAP and t-SNE, we clearly observed that three to five rare cells were mapped to the region of major cells (**left panel of Figure S7**).

In the second step of EDGE, we synergistically combine the second steps of UMAP and t-SNE. We choose to use UMAP’s distance distribution function for the low-dimensional embedding and the symmetrization method implemented in t-SNE as they are computationally efficient. To preserve global distances, we implement the loss function used in UMAP, see discussion on **Page 10** in Supplementary Information. Since the same loss function is used, the embedding result is analogous to the one constructed by the original UMAP algorithm if we use the similarity probabilities from UMAP to optimize the embedding in EDGE.

2. The evaluation of the simulation study is not direct. First, authors need to show the accuracy from random forest is a good evaluation metric. The embedding figures of the simulation study should be presented.

Response: Thank you for pointing it out. If the similarity learning strategy preserves within- and between-cluster distances successfully, various cell types should be mapped to

different compact regions in the low-dimensional space. Under this situation, nonlinear classifiers, such as random forests, have an excellent prediction performance in terms of classifying cells into different cell types (Breiman, 2001). To show this, we designed a simulation study using Splatter. Two settings with 10% and 5% differentially expressed (DE) genes were generated. The embedding results of EDGE were presented in Response Figure 1. We applied random forests to these two embedding matrices. For the setting of exclusive clusters, the prediction accuracy of random forests is 100%. For the overlapping setting, 16 cells were misclassified by random forests. The prediction accuracy reflects the degree of cluster separation. Thus, we applied random forests to evaluate whether cells were well-separated in the embedding space. Becht et al. (2019) also used random forests to evaluate the embedding performance of UMAP and t-SNE in the scRNA-seq data analysis. To fully address your concern, we have provided another metric, silhouette scores, to evaluate the performance of embeddings (Rousseeuw, 1987). The silhouette scores have been implemented to measure the degree of separation among clusters in the embedding space for the scRNA-seq data (Wu et al., 2018). The results of silhouette scores were consistent with the prediction accuracy based on random forests (**Figure 2**).

Response Figure 1: Embedding results of EDGE for exclusive and overlapping settings. Two types of cells are marked by red triangles and blue dots.

We repeated the simulation 100 times. As it would be impractical to show all embedding figures (27 scenarios \times 100 replications \times 3 methods) in the simulation, we presented two

embedding figures from Scenario 1 and 2 as examples in **Figure S12** in Supplementary Information. The embedding figures in Scenario 3 and 4 are similar to those in Scenario 1 and 2.

3. Both UMAP and tSNE don't always give consistent results from run to run. So comparison on one single run is not very convincing. I think authors should run multiple times and use a quantitative way to compare the results on both simulation data and real data.

Response: We agree with the reviewer that all three methods don't always give consistent results from run to run. Thus, we repeated the simulation and real data analysis **100** times and calculated prediction accuracy based on random forests and silhouette scores (**Figure 2 and highlighted sentences on Page 9 and 11 in the main article**).

Minor Comments:

4. I suggest authors use more comprehensive simulation study. Simulation study should include minor population ranging from 0.1% to 5%. Goal is to study the minimal detectable percentage of different methods. The number of clusters also have impact on the results. Authors should report how rare population detection accuracy change as the number of clusters increase (i.e. up 10 clusters).

Response: As suggested by the reviewer, we have provided additional simulation studies to investigate the behaviors of EDGE in detecting rare populations. We designed two additional scenarios, Scenario S1 and Scenario S2, in which the number of cells and genes were 10,000 and 500 respectively. The performance of EDGE, together with t-SNE and UMAP, was measured by the prediction accuracy of the rare population through out-of-bag (OOB) prediction errors over **100** simulation replicates. In **Scenario S1**, we let the total number of cell types be 5, among which one was a rare cell type with the percentage ranging from **0.1% to 5%**. The other four major cell types were set to have equal proportions. For settings with different rare cell percentages, EDGE achieved the highest prediction accuracy on average

(**Figure S8**). The median of prediction accuracy was above 80% for the rare cell group when the rare cell percentage was greater than 0.5%. To explore the performance of EDGE in detecting multiple rare populations, we further let the number of rare populations be three, and all other settings remained the same. EDGE still ranked first in terms of prediction accuracy in all three rare populations (**Figure S9**). In **Scenario S2**, we investigated the prediction accuracy of the rare population when the total number of cell types varied from **2 to 10**. Among all the cell types, one was set to be the rare population with a percentage of 1%. The major cell types were set to have equal proportions. In all settings, EDGE maintained the highest prediction accuracy on average when the number of cell types varied from 2 to 10 (**Figure S10**). We further let the number of rare populations be three, and the number of cell types varied from 4 to 10. All other settings remained the same. EDGE outperformed other methods consistently in all three rare populations (**Figure S11**).

5. The original paper of single cell brain data has detected nine clusters. But authors only compare 7 clusters. In addition, it seems the t-SNE in original paper has very good separation. I think authors should compare EDGE results with the t-SNE in original paper.

Response: Thank you for your suggestion. We downloaded the data from <http://linnarssonlab.org/cortex/>, which was the website for the original paper. Since all cell subtypes in Ependymal were presented in Astrocytes, these two cell types were combined by authors of the original paper, see the subtypes in mRNA raw data (https://storage.googleapis.com/linnarsson-lab-www-blobs/blobs/cortex/expression_mRNA_17-Aug-2014.txt). Mural and Endothelial cells were also combined by the authors due to the same reason. As we used the cell subtypes to illustrate how methods preserved the local and global structures, we followed the cell type merging procedure from the authors. We contacted an author of the original paper and received **partial details** about the implementation of t-SNE. Following these details, we changed the perplexity parameter to 50 and the theta parameter to 0.3 (**Figure 4**). There were seven large clusters (four cell types were merged into two cell

types) in our t-SNE figure. However, without further details such as the random seed and normalization method, we cannot regenerate the original figure.

6. The brain data is from Fluidigm C1, which data have much lower drop-out rate compared to drop-seq based methods. It seems EDGE can handle high drop-out sparse data. If this is the case, I think it is good to try this method on more challenging data like single cell chromatin accessibility (scATAC-seq). It's just a suggestion.

Response: We appreciate this valuable suggestion. Compared to scRNA-seq data, scATAC-seq data have two distinct features: higher dimensionality and higher sparsity (Buenrostro et al., 2015). For instance, in the recently published Splenocyte scATAT-seq dataset, there are about 80,000 features, and only 1-10% of expected accessible peaks are detected in the scATAT-seq dataset (Chen et al., 2019, 2018). This close-to-binary sparsity nature and high dimensionality make the **direct application** of EDGE, t-SNE, and UMAP to scATAC-seq data unsuitable (Xiong et al., 2019). One way to apply EDGE, t-SNE, and UMAP to the data is through a two-step procedure. We implemented this two-step procedure to the Forebrain scATAC-seq dataset published in Preissl et al. (2018). In the first step, the hidden feature detection method SCALE designed for the scATAC-seq data was implemented to extract low-dimensional features (Xiong et al., 2019). EDGE, t-SNE, and UMAP were applied to these hidden features in the second step. We presented the embedding results in Response Figure 2. The medians of silhouette scores for EDGE, UMAP, and t-SNE were 0.34, 0.32, and 0.28, respectively. Although EDGE has a better performance in embedding scATAC-seq data, the contribution of the embedding step is not clear as the hidden features are not sparse and are low-dimensional. This is an interesting topic. The direct application of EDGE in the raw scATAC-seq data deserves further investigation.

Response Figure 2: Embedding results of EDGE, UMAP, and t-SNE for the Forebrain scATAC-seq dataset. Eight types of cells are represented by dots with different colors.

Response to the Reviewer 2’s comments on “EDGE: Ensemble Dimensionality Reduction and Feature Gene Extraction for Single-cell RNA-seq Data”

We would like to thank you for your insightful comments and suggestions. We have made every effort to address the concerns raised by you and the other reviewer. We hope you find our responses satisfactory. In what follows, in italics we quote your comments and then give our detailed response. We showed all changes in the manuscript text file with color highlighting (blue).

Major Comments:

1. I found some of the method descriptions unclear. These should be addressed: i. A one sentence explanation beyond “similarity search method” on what a “sketch” is would be very helpful in the last paragraph on page 3. ii. What does a “vote” for cell similarity mean? iii. What exactly is the weak learner method? It is never described in the methods or in the supplement.

Response: Thank you for pointing this out. We appreciate this helpful advice. One sentence describing the ensemble learning method has been added on Page 2. **“Our method adopts an ensemble learning technique, which employs multiple weak learners to obtain better predictive performance than could be obtained from any weak learners alone (Hastie et al., 2009).”** We also discussed the ensemble learning method in Discussion on Page 14.

We have revised the caption of Figure 1 and the Similarity search section on **Page 14 and 15**. Below we provide a summary of revisions. Our algorithm starts with constructing a similarity learner. We randomly select a certain number of genes, e.g., B genes, for all cells. We then randomly pick a threshold for each gene within the range of all values in the gene expression matrix. **A sketch is a bit vector with B elements (0 or 1).** Each element is associated with a selected gene. If the gene expression value is greater than the gene’s threshold, its corresponding value in the bit vector is 1 and 0 otherwise. Cells share the

same bit vector are mapped to a hash code (imaginary box). The similarity scores of cells in the same hash code are set to be 1s. Once the pairwise similarity scores for all cells are calculated, the construction of a similarity learner is complete. **Since the thresholds and groups of genes are randomly selected, it is possible that the similarity score for dissimilar cells is 1. Thus, we call the similarity learner as the weak learner to reflect its accuracy in similarity learning.** To increase the accuracy of such similarity learning, we utilize ensemble learning by constructing a number of weak learners. **Each weak learner is a “voter”. The final similarity probabilities between cells are calculated by averaging the corresponding similarity scores from all “voters”.**

2. To justify the methods claim at preserving separation more accurately, the authors could vary the degree of separation among the clusters in the simulation and then define a measure of cluster distances and show whether the distance changes relatively for the EDGE method.

Response: This is an excellent point. Thank you for this helpful comment. For the scRNA-seq simulator Splatter, the data generation procedure consists of multiple stages. In each stage, different statistical distributions are used to simulate the parameters and data randomly. Thus, it is difficult to control and infer the degree of separation in the simulation precisely. However, the degree of separation is affected by the proportions of differentially expressed (DE) genes (**Response Figure 1 a, b**) and the number of major and rare cell types (**Figure S12 in Supplementary Information**). In our simulation studies, we varied the percentage of DE genes and the number of major and rare cell types to create scenarios with diverse degrees of separation. To estimate the degrees of separation among clusters in the embedding space, we further provided silhouette scores for each method (Rousseeuw, 1987). These scores have been used to evaluate the performance of embedding in the scRNA-seq data analysis (Wu et al., 2018). They measure how close a cell to other cells in its own cell type compared to cells in the next nearest cell type in the embedding space. Although we

cannot compare the true degree of separation with the estimated one, we have shown that EDGE has the highest silhouette scores on average in most settings (**Figure 2**).

Response Figure 1: Embedding results of EDGE for two simulated datasets. a: The proportion of DE gene is 10%. b: The proportion of DE gene is 5%. Two types of cells are marked by red triangles and blue dots.

3. *Unless the empirical datasets are both sorted single-cell populations, the methods should discuss how the ground truth labels were obtained originally as they were likely estimated using other statistical methods and so have some degree of uncertainty.*

Response: Thank you for pointing it out. We agree with you that the cell type labels are critical in real data analysis. We used three well-studied datasets in the paper. For the Jurkat dataset, high-quality single nucleotide variants (SNVs) observed only in each cell type were used to resolve the cell types. The PBMC cell types and associated expression patterns have also been well-studied. The cell types were identified using known marker genes. For the mouse brain dataset, cell types were resolved by using an algorithm developed by authors of the original paper. The results were further verified by biologists using known marker genes. We have revised the **Data availability** section on **Page 18 and 19** to review how labels of cell types have been obtained in each dataset.

4. *The package should be made available on GitHub with a full working vignette with*

examples. I was not able to test the package myself since there is no documentation on how to use the various functions.

Response: We appreciate this valuable suggestion. The package and working examples are available at <https://sites.google.com/view/xiaosun/edge>. We will make it **publicly** available on GitHub and R CRAN and continue to maintain and update the software in the future.

Minor Comments:

5. On page 5, “Diferentially expressed (DE) gene proportions for Scenario 1 to 3 were fixed at 35%,” diferentially should be differentially.

Response: Thank you for pointing it out. We have corrected it.

6. The grammar in Figure 1 caption can be cleaned up a bit as well as in the first methods paragraph.

Response: Thank you for your suggestion. We have revised the caption of Figure 1 and the first paragraph in Methods.

7. It would be more clear/fair to write the UMAP and TSNE were not designed to identify feature genes, in addition to saying they aren’t able to on Page 6 bottom paragraph.

Response: Thank you for your suggestion. We have corrected it.

References

- Becht, E., L. McInnes, J. Healy, C.-A. Dutertre, I. W. Kwok, L. G. Ng, F. Ginhoux, and E. W. Newell (2019). Dimensionality reduction for visualizing single-cell data using UMAP. *Nature Biotechnology* 37(1), 38.
- Breiman, L. (2001). Random forests. *Machine Learning* 45(1), 5–32.
- Buenrostro, J. D., B. Wu, U. M. Litzénburger, D. Ruff, M. L. Gonzales, M. P. Snyder, H. Y. Chang, and W. J. Greenleaf (2015). Single-cell chromatin accessibility reveals principles of regulatory variation. *Nature* 523(7561), 486–490.
- Chen, H., C. Lareau, T. Andreani, M. E. Vinyard, S. P. Garcia, K. Clement, M. A. Andrade-Navarro, J. D. Buenrostro, and L. Pinello (2019). Assessment of computational methods for the analysis of single-cell ATAC-seq data. *Genome Biology* 20(1), 1–25.
- Chen, X., R. J. Miragaia, K. N. Natarajan, and S. A. Teichmann (2018). A rapid and robust method for single cell chromatin accessibility profiling. *Nature Communications* 9(1), 1–9.
- Hastie, T., R. Tibshirani, and J. Friedman (2009). *The elements of statistical learning: data mining, inference, and prediction*. Springer Science & Business Media.
- Preissl, S., R. Fang, H. Huang, Y. Zhao, R. Raviram, D. U. Gorkin, Y. Zhang, B. C. Sos, V. Afzal, D. E. Dickel, et al. (2018). Single-nucleus analysis of accessible chromatin in developing mouse forebrain reveals cell-type-specific transcriptional regulation. *Nature Neuroscience* 21(3), 432–439.
- Rousseeuw, P. J. (1987). Silhouettes: a graphical aid to the interpretation and validation of cluster analysis. *Journal of Computational and Applied Mathematics* 20, 53–65.
- Wu, Y., P. Tamayo, and K. Zhang (2018). Visualizing and interpreting single-cell gene expression datasets with similarity weighted nonnegative embedding. *Cell Systems* 7(6), 656–666.

Xiong, L., K. Xu, K. Tian, Y. Shao, L. Tang, G. Gao, M. Zhang, T. Jiang, and Q. C. Zhang
(2019). SCALE method for single-cell ATAC-seq analysis via latent feature extraction.
Nature Communications 10(1), 1–10.

REVIEWERS' COMMENTS:

Reviewer #1 (Remarks to the Author):

The authors solved all my comments. I have no further comments.

Reviewer #2 (Remarks to the Author):

The authors have satisfied and addressed all comments.

Only additional comment is to make sure the text and labels in all figures is a bit larger in the final version for publication. The rendered PDF makes the figure text appear quite small.

Response to the Reviewer 1's comments on "Ensemble Dimensionality Reduction and Feature Gene Extraction for Single-cell RNA-seq Data"

- 1. The authors solved all my comments. I have no further comments.*

Response: Thank you for your helpful suggestions and comments!

Response to the Reviewer 2's comments on "Ensemble Dimensionality Reduction and Feature Gene Extraction for Single-cell RNA-seq Data"

- 1. The authors have satisfied and addressed all comments. Only additional comment is to make sure the text and labels in all figures is a bit larger in the final version for publication. The rendered PDF makes the figure text appear quite small.*

Response: Thank you for your helpful suggestions and comments! To make text and labels easier to read, we have increased the font size of them in all figures.